# Probiotics as Potential Tool to Mitigate Nucleotide Metabolism Alterations Induced by DiNP Dietary Exposure in *Danio rerio*

**DOI:** 10.3390/ijms252011151

**Published:** 2024-10-17

**Authors:** Christian Giommi, Francesca Maradonna, Claudia Ladisa, Hamid R. Habibi, Oliana Carnevali

**Affiliations:** 1Department of Life and Environmental Sciences, Polytechnic University of Marche, 60131 Ancona, Italy; c.giommi@staff.univpm.it (C.G.); f.maradonna@staff.univpm.it (F.M.); 2INBB—Biostructures and Biosystems National Institute, 00136 Roma, Italy; 3Department of Biological Sciences, University of Calgary, Calgary, AB T2N 1N4, Canada; cladisa@verschurencentre.ca (C.L.); habibi@ucalgary.ca (H.R.H.)

**Keywords:** SLAB51, phthalates, nucleotide metabolism, endocrine disruptors, hepatotoxicity, zebrafish

## Abstract

Diisononyl phthalate, classified as endocrine disruptor, has been investigate to trigger lipid biosynthesis in both mammalian and teleostean animal models. Despite this, little is known about the effects of DiNP exposure at tolerable daily intake level and the possible mechanisms of its toxicity. Probiotics, on the other hand, were demonstrated to have beneficial effects on the organism’s metabolism and recently emerged as a possible tool to mitigate the EDC toxicity. In the present study, using a metabolomic approach, the potential hepatic sex-related toxicity of DiNP was investigated in adult zebrafish together with the mitigating action of the probiotic formulation SLAB51, which has already demonstrated its ability to ameliorate gastrointestinal pathologies in animals including humans. Zebrafish were exposed for 28 days to 50 µg/kg body weight (bw)/day of DiNP (DiNP) through their diet and treated with 10^9^ CFU/g bw of SLAB51 (P) and the combination of DiNP and SLAB51 (DiNP + P), and the results were compared to those of an untreated control group (C). DiNP reduced AMP, IMP, and GMP in the purine metabolism, while such alterations were not observed in the DiNP + P group, for which the phenotype overlapped that of C fish. In addition, in male, DiNP reduced UMP and CMP levels in the pyrimidine metabolism, while the co-administration of probiotic shifted the DiNP + P metabolic phenotype toward that of P male and closed to C male, suggesting the beneficial effects of probiotics also in male fish. Overall, these results provide the first evidence of the disruptive actions of DiNP on hepatic nucleotide metabolism and mitigating action of the probiotic to reduce a DiNP-induced response in a sex-related manner.

## 1. Introduction

Diisononyl phthalate (DiNP), a plasticizer belonging to the phthalate ester family, is commonly used to confer flexibility, durability, and resilience to polyvinyl chloride (PVC) and other polymers. It is a component of various consumer products, including food packaging and medical devices and has been introduced to replace some previously banned phthalates, including Di-(2-ethylhexyl) phthalate (DEHP), for which hepatic toxicity has been largely documented [1,2,3,4]. A great deal of evidence shows that DiNP can leach out and enter the body, being detected in different biofluids such as blood and urine [5,6]. The observed toxic effects of DEHP range from the induction of nonalcoholic fatty liver disease (NAFLD) [7,8,9,10] to the impairment of liver energy production pathways, coupled with the increased purine catabolism [11]. Although DiNP is considered a safer alternative to DEHP, concerns have emerged regarding its capacity to affect human health by promoting lipid de novo synthesis through the interaction with peroxisome proliferator receptors (PPARs) [12]. This last finding is consistent with previous studies demonstrating the DiNP-induced onset of liver steatosis in teleost [13,14,15] and mammals [16]. Furthermore, hepatic damage has been observed in seabream fed a diet contaminated with DiNP [13,17] and in mice, where DiNP exposure induced liver oxidative stress and caused histological alterations [18]. In addition, in zebrafish, further evidence has demonstrated the disruption of reproduction following exposure to DiNP [19,20,21]. However, no consensus exists regarding the mode of action through which DiNP exerts its endocrine disruptive activity. In fact, while using an estrogen-responsive ChgH-EGFP transgenic model of medaka eleutheroembryos [22], DiNP has been demonstrated to possess estrogenic activity, on the countrary, by using a XenoScreen YES/YAS in vitro assay, it was shown that DiNP does not possess estrogenic or androgenic action, instead revealing the pollutants anti-estrogenic and anti-androgenic properties [23]. Although there are contrasting results regarding this phthalate hormonal properties, its role as an endocrine-disrupting chemical (EDC) is nevertheless evident, and the questioning of the suitability of DiNP as a safe DEHP substitute is increasing. As result, in 2019, the European Food Safety Authority (EFSA) reduced the tolerable daily intake (TDI) of DiNP to a limit of 50 µg/kg bw/day [24], but unfortunately, it is still widely present in commonly used plastic products and in the environment, thus increasing the concern regarding its adverse health impact. Unless we see a significant decrease in the production and use of various substances capable of disrupting the endocrine system, new strategies to mitigate the harmful effects of environmental contaminants are needed. Among these strategies, we come across natural and synthetic antioxidants ranging from polyphenols, catechins, and vitamins [25,26,27,28,29,30,31], or natural products with positive effects on health [32] such as *Glycyrrhiza glabra* root extract [33] and mulberry crude extract [34]. Over the years, studies have demonstrated the ability of probiotics to improve metabolic health. In this regard, in zebrafish, the administration of *Lactobacillus rhamnosus* reduced liver oxidative stress, DNA damage, and apoptosis [35], and, in Sprague Dawley rats, the administration of *Bacillus SC06* before the induction of oxidative stress alleviated liver injuries by increasing the liver antioxidant capacity [36]. This evidence suggests the possible use of probiotics to protect the organism against DiNP detrimental effects. In this contest, numerous indications so far have proved the effectiveness of probiotics in reducing the symptoms of liver steatosis and NAFLD [37,38,39,40]. In addition, evidence of the ability of probiotics to mitigate EDC toxicity on reproductive [41], metabolic [42,43,44,45], neurological [43,46], and immune system [43,47] functions exists. Furthermore, there is evidence that the probiotic formulation SLAB51 reduces neurodegenerative disorders [48] and gastrointestinal pathologies [49,50,51] and mitigates disruptive actions of Bisphenol A (BPA) on the gut microbiota–liver–brain axis in zebrafish [43]. Only a few studies are available regarding the toxicity exerted by DiNP on hepatic metabolism. In the present study, starting from the already existing information regarding DiNP hepatic toxicity [13,14,16,52], we used a metabolomic approach to more deeply investigate hepatic metabolic pathways impacted by this contaminant exposure and explore the mitigating effects of SLAB51 on DiNP metabolic disturbance in both male and female zebrafish.

## 2. Results

### 2.1. Food Intake

During the trial, food intake did not change among experimental groups. Fish from all the experimental groups consumed the daily food (1.44 g; 3% of body weight) within 30 min of the analysis. Since 100% of the daily administered food was consumed by the fish from all the experimental groups, the remains could not be collected from the tank.

### 2.2. Metabolomic Analysis

#### 2.2.1. Metabolomic Characterization of Liver through PCA and PLS-DA Analysis

To analyze the hepatic metabolic changes in the different experimental groups, intensity peaks generated from UHPLC-MS were selected in MAVEN software version 3.9.9 coupled with a standard library of the mass to charge (*m*/*z*) and retention times of metabolites, finding 71 metabolites in total. The dataset produced was then analyzed using a combination of multivariate analysis and visualization techniques such as PCA and PLS-DA. Regarding PCA, the quality control group (QC, *n* = 4), comprising seven extracted pools of female livers mixed with four extracted pools of male livers, were included in the analysis in order to test the goodness and reproducibility of the analysis. PCA demonstrated a strong cluster formation regarding the QC samples in both sexes (Appendix A); however, a clear separation among experimental groups was not evident with this analysis. To better investigate differences among groups in both sexes, PLS-DA analyses were conducted using the SIMCA program, analyzing all groups together and in a pairwise manner (Figure 1 and Figure 2), with R2 and Q2 used to assess the quality of the built models reported for each tested comparison. Concerning females, the comparison of all groups together revealed that the group DiNP + P overlapped with the C group (Figure 1a), while P and DiNP were clustered separately. The pairwise comparison of groups in females revealed that it was consistent with the PLS-DA of all groups; only the comparison of C and DiNP + P showed a partial overlap regarding Component 1 (Figure 1c), while the other comparison evidenced a clear separation. In males, in the comparison of all groups, the DiNP + P group was clustered with the P group, and both were clustered closer to the C group, showing no separation in Component 2 between C, P, and DiNP + P. Interestingly no overlap was observed among DiNP and the other groups (Figure 2a), highlighting that only in this last group was a significant altered hepatic metabolic profile induced. In males, the pairwise comparison among all other groups revealed a lack of clear separation between C and DiNP + P in terms of Component 1 of the model (Figure 2c). However, the other groups clustered distinctly in relation to Component 1 of the models, indicating that in the overall comparison, P and DiNP + P may not be clearly distinguished (Figure 2f) with respect to the first two components. However, they may potentially be distinguished with respect to a third component, which appears to be predominant in the P vs. DiNP + P comparison. For a more comprehensive investigation, changes observed at the metabolic level due to the treatments in all these comparisons were examined. Specifically, metabolites with a Variable Important in Projection (VIP) score of ≥1 from all the constructed models were selected (see Appendix A) and utilized in the following univariate analyses.

#### 2.2.2. Differential Metabolite Analysis through Volcano Plot

The selected VIPs for each PLS-DA model were used for univariate analyses to investigate changes induced by each treatment against the C using Metaboanalyst 5.0 software. For each comparison, data were analyzed using a volcano plot, which considered both the fold change (FC) [log2(FC)] and *p*-value [−log10(p)] (Figure 3, Figure 4 and Figure 5). The volcano plot results are also summarized in Appendix A.

In females, DiNP alone induced a decrease in guanosine monophosphate (GMP), adenine monophosphate (AMP), and inosinic acid (IMP) and an increase in hippuric acid compared to C (Figure 3a). The co-treatment of DiNP and SLAB51 led to the increase in hypotaurine when compared to C (Figure 3b), while P treatment showed a reduction in AMP and increaseds level of sucrose, uric acid (UA), and 3-hydroxybutiric acid with respect to C (Figure 3c).

In males, the exposure to DiNP decreased pyridoxal, uridine 5′-mnophosphate (UMP), and cytidine monophosphate (CMP) accompanied by a concurrent increase in N-acetyl-L-methionine, L-methionine, L-alanine, glutaric acid, and xanthine with respect to C (Figure 4a). When administered together, DiNP and SLAB51 induced an increase in docosahexaenoic acid (DHA) and hypoxanthine compared to C (Figure 4b). Conversely, when administered alone, P caused a reduction in pyridoxal and N-acetylleucine, alongside an increase in N-acetyl-L-methionine, DHA, uridine diphosphate-N-acetylglucosamine, and L-alanine compared to the C group (Figure 4c).

A subsequent univariate analysis was conducted to investigate changes in the metabolic profile of treatments by comparing DiNP + P singularly with DiNP and P in both sexes. In females, the comparison of DiNP + P vs. DiNP resulted in a decrease in D-ribose levels together with an increase in hipotaurine and ophthalmic acid in DiNP + P (Figure 5a), while in comparing DiNP + P vs. P, the co-treatment induced a reduction in panthotenic acid and an increase in hypotaurine (Figure 5c). In males, the DiNP + P group exhibited a decrease in L-arginine, L-tyrosine, L-phenylalanine, L-methionine, L-cystathionine, L-glutamine, diaminopimelic acid, diidroxyacetone phosphate, L-isoleucine, and xanthine and an increase in betaine and pyridoxal compared to the DiNP group (Figure 5b). The comparison of DiNP + P with P resulted in a decrease in diidroxyacetone phosphate and an increase in betaine, D-ribose, and hypotaurine in DiNP + P with respect to P (Figure 5d).

#### 2.2.3. Pathway Analysis

To explore the effects of the metabolic changes observed among the various groups, a comprehensive metabolic pathway analysis (MetPA) was carried out. This involved plotting the impact of each metabolite within the inferred pathway alongside the corresponding *p*-value [−log10(p)] (see Figure 6 and Figure 7). The results of the MetPA analysis are further detailed in Appendix A.

In females, MetPA analysis pinpointed the impact on purine metabolism, pyrimidine metabolism, pentose and glucuronate interconversion, amino sugar and nucleotide sugar metabolism, and ascorbate and aldarate metabolism in DiNP compared to C (Figure 6a).

The pathway analysis conducted on C vs. DiNP + P found an impact on taurine and hypotaurine metabolism, purine metabolism, and aminoacyl-tRNA biosynthesis (Figure 6c).

The comparison of C vs. P showed that galactose metabolism, starch and sucrose metabolism, fructose and mannose metabolism, glycolysis/gluconeogenesis, and the synthesis and degradation of ketone bodies are the most impacted pathways (Figure 6e).

In males, the comparison of C and DiNP by MetPA analysis revealed the impact on vitamin B6 metabolism; aminoacyl-tRNA biosynthesis; cysteine and methionine metabolism; pyrimidine metabolism; alanine, aspartate, and glutamate metabolism; and seleno compound metabolism (Figure 6b). When comparing C with DiNP + P, pathway analysis suggested the impact on pyruvate metabolism, arginine biosynthesis, citrate cycle (TCA cycle), the biosynthesis of unsaturated fatty acid, d-glutamine and d-glutamate metabolism, vitamin B6 metabolism, and purine metabolism (Figure 6d).

Lastly, the comparison between C and P highlighted the impact on histidine-, glyoxylate, and dicarboxylate metabolism; arginine biosynthesis; and d-glutamine and d-glutamate metabolism, as well as vitamin B6 metabolism (Figure 6f).

MetPA analysis was also conducted to explore the impact of treatments on metabolic pathways by comparing DiNP + P with DiNP and P in a pairwise manner for both sexes. In females, when comparing DiNP + P vs. DiNP, MetPA revealed the impact on taurine and hypotaurine metabolism; alanine, asparagine, and glutamate metabolism; citrate cycle (TCA cycle); pyruvate metabolism; and nicotinate and nicotinamide metabolism (Figure 7a). On the other hand, compared to P, DiNP + P exhibited notable impact on galactose metabolism, starch and sucrose metabolism, amino sugar and nucleotide sugar metabolism, citrate cycle (TCA cycle), pyruvate metabolism, and pantothenate and CoA biosynthesis (Figure 7c).

In males, when comparing DiNP + P with DiNP, the results reveal an impact on arginine and proline metabolism; phenylalanine metabolism; phenylalanine, tyrosine, and tryptophan biosynthesis; histidine metabolism; and beta-alanine metabolism (Figure 7b). In the comparison of DiNP + P vs. P, the most noticeable impact was found on glycine, serine, and threonine metabolism; arginine and proline metabolism; glutathione metabolism; the pentose phosphate pathway; inositol phosphate metabolism; and taurine and hypotaurine metabolism (Figure 7d).

## 3. Discussion

In recent years, metabolomics has emerged as cutting-edge technology and a powerful analytical tool to advance the knowledge on toxicological studies, able to provide comprehensive insights into the dynamic and intricate metabolic changes induced by toxic substances [53,54,55]. Moreover, considering that approximately 70% of human genes have at least one obvious zebrafish orthologue [56] and due to the high similarity in liver functionality compared to humans [57,58,59,60], zebrafish’s importance as an animal model for biomedical research [61,62,63] and toxicological studies [64,65] has increased significantly in recent decades. Xenobiotics, indeed, including phthalates, enter the body through different routes, triggering complex metabolic processes essential for their neutralization and subsequent elimination. In this regard, with its remarkable detoxification capabilities, the liver emerges as a central hub for these critical functions [66,67]. The correct functioning of the liver is also critical in the regulation of different metabolic pathways, including nucleotide metabolism, which is pivotal in sustaining RNA, DNA, membrane lipid synthesis, and the protein glycosylation process [68,69,70]. Nucleotide metabolism predominantly occurs within the cytoplasm of hepatocytes, initiated by glucose conversion into D-ribose. Subsequently, D-ribose, via the pentose phosphate pathway, is converted into phosphoribosyl pyrophosphate (PRPP), the fundamental precursor for all nucleotide intermediates [69]. *Danio rerio* and humans present a high similarity regarding the nucleotide metabolism and its intermediate molecules, with zebrafish being currently considered a suitable model for the investigation of nucleotide metabolism disorders [71]. The results obtained in the present study, for the first time, provide evidence regarding the capacity of DiNP to alter hepatic nucleotide metabolism, a toxic outcome so far described for phthalates other than DiNP in vivo and in vitro models. In crucian carp (*Carassius auratus*) chronically exposed to DEHP, an enhanced hepatic purine metabolism was observed [11], while in human liver cancer cells (HepG2) exposed to dibutyl phthalate (DBP), a decrease in purine metabolism intermediates such as AMP, IMP, GMP, and dGMP was measured, where this effect intensified in the case of co-exposure with DEHP [72]. In addition, our results also pinpoint the sex-specific modulation of nucleotide metabolism: in females, DiNP altered purine metabolism, as indicated by the down-regulation of all purine metabolism intermediates. In this contest, previous in vitro studies have demonstrated the capacity of DiNP to induce DNA synthesis in mouse hepatic cells [73], along with its ability to modify the expression of genes involved in DNA repair and recombination [12], suggesting an augmented demand for nucleic acid synthesis after DiNP-induced toxicity and hepatocyte proliferation. Moreover, the role of nucleotides on lipid accumulation and adipogenesis recently emerged in vitro, showing that nucleotide biosynthesis inhibition impairs lipid storage and down-regulates the expression of lipogenic factors [74]. Conversely, the administration of purine to cultured *Oncorhynchus mykiss* hepatocytes fostered lipid biosynthesis [75]. In this light, the enhanced catabolism of purine intermediates could also represent a consequence of the well-known ability of DiNP to induce lipid biosynthesis and act as a possible obesogenic. PPARs act as central regulators of lipid metabolism [76] and energy homeostasis [77]. Their dysregulation consequently led to dyslipidaemia and metabolic diseases. The ability of DiNP to interact with PPAR is well known [12,73] and is in line with previous results obtained in our laboratory demonstrating the disruption of hepatic lipid metabolism in different teleost species exposed to DiNP [13,14,15] and in mammals [16,78,79]. Noteworthy, in females, an increase in hippuric acid was also found, and since this metabolite is a catabolite of food-derived benzoic acid [80], its presence suggests the onset of aciduria, a condition that predisposes to diabetes [81]. Interestingly, in females, the co-treatment of DiNP with SLAB51 counteracted the observed DiNP-induced decrease in purine intermediates, probably using D-ribose as substrate for their production. This assumption is based on the reduction in D-ribose observed in DiNP + P fish compared to DiNP ones. DiNP can also trigger hepatic oxidative stress [18,82,83], in both teleost and mammals. Interestingly, the observed increase in hypotaurine, an antioxidant metabolite [84], in DiNP and SLAB51 co-exposed females suggests that the probiotic administration enhances the antioxidant capacity of the liver to face the oxidative stress caused by DiNP. The rise in this metabolite could also be responsible for the reduction in liver aciduria found in this group, associated with the reduction in hippuric acid. The beneficial role of the probiotic mix herein used, is also supported by the analysis of the metabolic phenotype, which highlights differences between the C and DiNP groups and the overlap between the C and DiNP + P groups, indicating a complete mitigation of metabolic toxicity. Similarly, some phthalate esters were observed to interfere with pyrimidine metabolism. Exposing HepG2 cells to DBP and its combination with DEHP increased the pyrimidine intermediates [72]. Furthermore, in patients with type 2 diabetes mellitus, CMP serum levels were found to be positively correlated with the urinary levels of DEHP [85]. On the contrary, a decreased cytosine level was observed in the breast cancer cell line MCF-7 exposed to DEHP [86], as well as a decreased level of UMP in trophoblast cell line HTR-8/Svneo exposed to Mono(2-ethylhexyl) phthalate (MEHP), a DEHP metabolite [87]. Regarding males, our present observation suggests that DiNP exposure mainly impacts pyrimidine metabolism through decreasing UMP and CMP levels. These metabolites could be catabolized to sustain hepatic metabolic activity. This hypothesis agrees with the recently discovered role of UMP in regulating glucose, lipid, and amino acid homeostasis [88]. This metabolite is indeed used as an energy source for the de novo synthesis of lipids, as demonstrated in a study following the administration of UMP during early weaning in piglets [89]. Furthermore, in DiNP-exposed male, the observed increase in xanthine, a final product of purine metabolism [90], suggests a possible impact of DiNP on this pathway. The hepatic metabolic disturbance induced by DiNP was also confirmed in this group by the observed increased levels of the amino acid L-alanine, glutaric acid (a catabolite product), L-methionine, and its acetylated form, N-acetyl-L-Methionine, suggesting an enhancement of amino acid metabolism. All these metabolites could potentially have harmful effects. Specifically, L-methionine accumulation could result in cardiotoxic effects outside the liver [91], while the increase in glutaric acid can be associated with the onset of aciduria, which can cause, in the long term, brain damage [92]. The decrease in this group of pyridoxal metabolites also indicates the alteration of hepatic metabolic activity induced by DiNP. Being the active form of vitamin B6, it is required as a coenzyme for more than 150 biochemical reactions, such as the metabolism of carbohydrates, lipids, amino acids, and nucleic acids and also serves as an antioxidant [93]. Considering this, the decreased level of pyridoxal suggests a reduction in antioxidant defense and an increased energetic requirement. In addition, pyridoxal is also required by the alanine transaminase (ALT), a critical hepatic enzyme used as a marker of liver health [94] and as a coenzyme for the production of L-alanine and α-ketoglutarate [95]. The reduction in pyridoxal and the reduction in pyridoxal-alanine in males exposed to DiNP suggest the ability of DiNP to affect hepatic metabolism, interfering with ALT function. None of these metabolic changes were found when probiotic was co-administered with DiNP in males, except for the increase in hypoxanthine, a xanthine precursor, which, so far, is not related to any metabolic toxicity. However, the metabolic phenotype showed only partial mitigation exerted by SLAB51, as evidenced by PLS-DA analysis. The observed partial mitigation of DiNP by probiotics in males may be mediated by decreased L-methionine and xanthine levels, concomitant with increased pyridoxal levels, suggesting a recovery of ALT functionality. Concomitantly, the co-treatment DiNP + P increased hepatic betaine levels, which is an amino acid derivative known to be able to counteract metabolic disturbances [96,97,98,99] and is considered a marker of good metabolic functionality. This metabolite can prevent and be used to treat fatty liver disease in mice, potentially reversing insulin resistance [100] and counteracting hepatic oxidative stress by affecting liver mitochondrial function [101]. The elevation of betaine levels in the DiNP + P compared to the DiNP group may, in part, mitigate the toxic effects of phthalate exposure and potentially enhance hepatic metabolism.

The results evidenced the presence of a sex-specific toxicity of DiNP that could be possibly explained by the known hormone-like capacity of this contaminant. This idea is supported by a previous study describing the sex-specificity of zebrafish hepatic metabolism [102], either in physiological conditions or when treated with sex steroids, indicating that females are relatively insensitive compared to males, which were readily affected. Considering this evidence and that DiNP can act as an anti-androgenic compound [23], it is worth noting that hormonal regulation can influence various metabolic pathways and thus justify the sex differences observed here. Indeed, females were less impacted than males, and as a result, SLAB51 in males could only partially mitigate DiNP toxicity. Deeper studies into the interplay between different metabolic pathways and hormone signaling are requested to identify new connections between the androgen receptor and nucleotide metabolism.

Although the results obtained clearly meet the main aim of this study, there were some unavoidable limitations. Noteworthy, the metabolomic profiles provide a clear snapshot of hepatic changes occurring following the different dietary regimens, but a deeper investigation could help in understanding the pathways leading to the different phenotypes. Based on our results, we can infer that hepatic changes may reflect potential intestinal disturbances caused by the various treatments. Unfortunately, the intestine was not included in our study, but future research should prioritize investigating potential alterations at the gut level.

## 4. Materials and Methods

### 4.1. SLAB51 Probiotic Formulation

SLAB51 (SivoMixx^®^, Ormendes SA, Jouxtens-Mézery, CH, Switzerland) is a commercial probiotic formulation containing eight lyophilized bacterial strains [*Streptococcus thermophilus* DSM 32245 (80 billion CFU), *Bifidobacterium lactis* DSM 32246 (25 billion CFU), *Bifidobacterium lactis* DSM 32247 (25 billion CFU), *Lactobacillus acidophilus* DSM 32241 (5 billion CFU), *Lactobacillus helveticus* DSM 32242 (1 billion CFU), *Lactobacillus paracasei* DSM 32243 (12 billion CFU), *Lactobacillus plantarum* DSM 32244 (16 billion CFU), *Lactobacillus brevis* DSM 27961 (36 billion CFU)] with a total concentration of 2 × 10^11^ CFU/g.

### 4.2. Fish Maintenance

The experiment was carried out on six-month-old male and female *Danio rerio* (AB strain). Fish were reared under controlled conditions (28.0 ± 0.5 °C and 14/10 h of light/dark photoperiod). The chemical and physical water parameters were constantly monitored, and fish were fed daily with a quantity of dry food representing 3% of their body weight.

### 4.3. Chemical Exposure

DiNP (purity ≥ 99%) was purchased from Sigma-Aldrich (Oakville, ON, Canada). A stock solution was prepared by dissolving DiNP in acetone. A working solution was then prepared and sprayed on the fish food (TetraMin Granules; Tetra, Melle, Germany) with a concentration of acetone (vehicle) of 0.01% (*v*/*v*), which has already been demonstrated not to be toxic for zebrafish adults [103]. The food was then left to dry at room temperature overnight. Fish food for the C and P groups was prepared by spraying the same amount of acetone used for the preparation of DiNP and DiNP + P group’s diets on the food and was then left to dry at room temperature overnight.

### 4.4. Experimental Design

Fish were equally divided into 4 groups in triplicate in 30 L glass tanks, each containing 16 fish, maintaining a 1:1 sex ratio and divided as follows: control group (C), fed with commercial dry food (TetraMin Granules; Tetra, Melle, Germany) + vehicle; SLAB51 group (P), probiotic-treated group receiving commercial dry food + vehicle and, after drying, enriched with lyophilized probiotic SLAB51 (SivoMixx, Ormendes SA, Jouxtens-Mézery, CH, Switzerland) at a final concentration of 10^9^ CFU/g of bw; diisononylphtalate group (DiNP), fed commercial diet contaminated with 50 µg/kg bw/day DiNP + vehicle; and diisononylphtalate + SLAB51 (DiNP + P), fed commercial diet contaminated with 50 µg/kg bw/day DiNP + vehicle and, after drying, enriched with lyophilized probiotic SLAB51 (SivoMixx, Ormendes SA, Jouxtens-Mézery, CH, Switzerland) at a final concentration of 10^9^ CFU/g of bw. The DiNP dose in the food was the tolerable daily intake (TDI) established by EFSA in 2019 [24]. The SLAB51 concentration was selected based on previous studies on zebrafish [41,43] and mice [104,105].

### 4.5. Ethic Statement

All the procedures involving animal exposure and manipulation were conducted according to the University of Calgary animal care protocol (AC15-0183) for the care and use of experimental animals, and all efforts were adopted to minimize suffering. The trial lasted for 28 days, at the end of which the fish were euthanized using an excess of MS-222 (3-aminobenzoic acid ethyl ester; Sigma Aldrich, St. Louis, MO, USA) buffered to pH 7.4, as indicated by the animal care protocol. Liver samples were collected for metabolomic analysis and stored at −80 °C until further processing.

### 4.6. Food Intake (FI)

Before the FI test, the tanks were siphoned and cleaned to avoid the presence of excrement during the test, and water was replaced. Fish received pre-weighted food (3% of body weight) daily, and FI was calculated after 30 min on the total amount of food administered daily, as FI = Wi − Wf, where Wi = initial dry food weight and Wf = remaining dry food left after 30 min. FI analysis was performed daily the last five days of treatment.

### 4.7. Metabolite Extraction and UHPLC-ESI-MS Analysis

The extraction, separation, and identification of liver metabolites were performed as previously described [43] and are here briefly reported. Livers were pooled from at least 3 animals (at least ~25 mg per pool), each representing *n* = 1 for all groups. Replicates included *n* = 7 for females and *n* = 4 for males for each experimental group. The size of the replicates was determined by the weight of the livers, smaller in males than in females. Quality controls (QC, *n* = 4) were generated by pooling random samples and were also included in the analysis for the identification of possible outliers and to assess the dataset’s reliability. Metabolite separation was performed using Ultra-High-Performance Liquid Chromatography (UHPLC) coupled with high-resolution full-scan Mass Spectroscopy (MS) in negative-mode electrospray ionization for MS spectra acquisition. Metabolite identification was then carried out using a standard library of mass to charge (*m*/*z*), and the retention times and area top to measure peak intensity were analyzed using MAVEN freeware version 3.9.9.

### 4.8. Statistical Analysis

#### 4.8.1. Metabolomic Statistical Analysis

SIMCA (Umetrics, Umeå, Sweden) software version 14.0 was used to perform unsupervised Principal Component Analysis (PCA) of all treatments and QC. In addition, Partial Least Squares Discriminant Analysis (PLS-DA) was conducted, and the quality assessment of the models was indicated by R2 and Q2 > 0.5 [43,106,107]). Univariate analysis was performed using a volcano plot on metabolites with a VIP (Variable Importance in Projection) score > 1 for each model using Metaboanalyst version 5.0 (accession date 18 September 2023) as described previously [43,106,107]. A significant threshold of *p*-value < 0.05 coupled with a fold change > 2 was used to statistically assess significant differences among experimental groups.

#### 4.8.2. MetPA Pathway Analysis

Metabolites with a VIP score > 1 found in each PLS-DA model were used to perform pathway analysis (MetPA) using the Metaboanalyst 5.0 platform [43,106,107]. In this technique, the concentration of each metabolite [using Quantitative Enrichment Analysis (QEA)] and the position in the pathway (using Topological Analysis) were considered. The global test and relative-betweenness centrality algorithms were selected for these two parameters, and the KEGG pathway library of zebrafish (*Danio rerio*) was used as a reference.

## 5. Conclusions

Our study revealed a sex-related metabolomic dysregulation following exposure to DiNP, a finding of significant interest due to the abundance of this contaminant in the environment. The metabolomic analysis showed the adverse impact of DiNP exposure on purine metabolism in females and pyrimidine metabolism in males. Bibliographical evidence shows the weak estrogenic properties of DiNP, while its antiandrogenicity was documented. Our results suggest a sex-specific effect on nucleotide synthesis, but so far, evidence linking these results to the hormone-like capacity of the compound is lacking. The co-administration of SLAB51 mitigates the metabolic phenotype perturbance induced by DiNP in females, while in males, a partial mitigation was achieved. This study confirms the validity of the selected probiotic formulation in the mitigation of DiNP detrimental effects, suggesting its potential use as a tool to counteract the metabolic disturbance of this contaminant. Given the high homology between the zebrafish and human genome, these findings raise important questions about the potential toxicity of DiNP to human health and on the beneficial effects of probiotic use to counteract EDC toxicity. In conclusion, our results demonstrate that exposure to DiNP at a concentration established for TDI by EFSA leads to metabolic perturbance in the liver of zebrafish. The evidence reported in the present study may represent a starting point for additional studies on the possible toxicity of DiNP at the hepatic level and in other organs, with the aim to deeply describe this contaminant detrimental effects to help policy makers re-evaluate its limits.

## Figures and Tables

**Figure 1 ijms-25-11151-f001:**
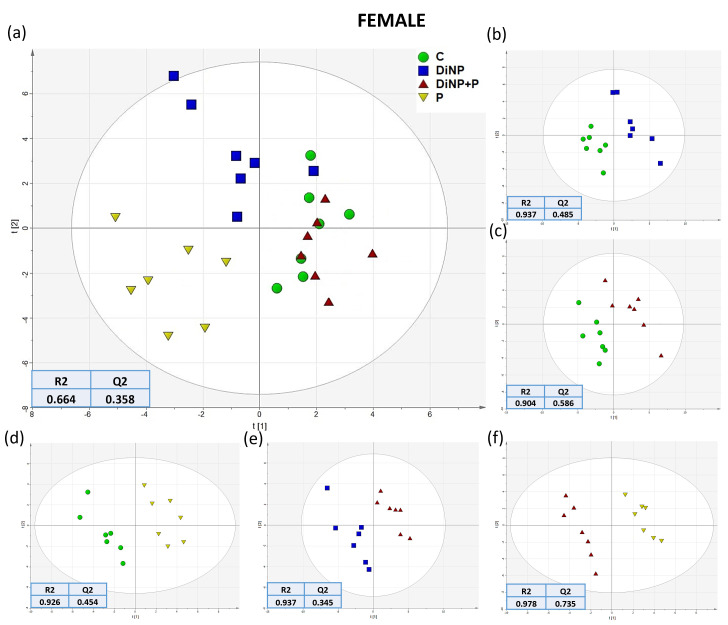
PLS-DA score plot analysis of (**a**) all groups, (**b**) DiNP vs. C, (**c**) DiNP + P vs. C, (**d**) P vs. C, (**e**) DiNP + P vs. DiNP, and (**f**) DiNP + P vs. P comparisons in female (*n* = 7 per group) with C (green), DiNP (blue), DiNP + P (red), and P (yellow). Each symbol in the PLS-DA represents a pool of at least three livers. Quality parameters of the built models are reported for each comparison.

**Figure 2 ijms-25-11151-f002:**
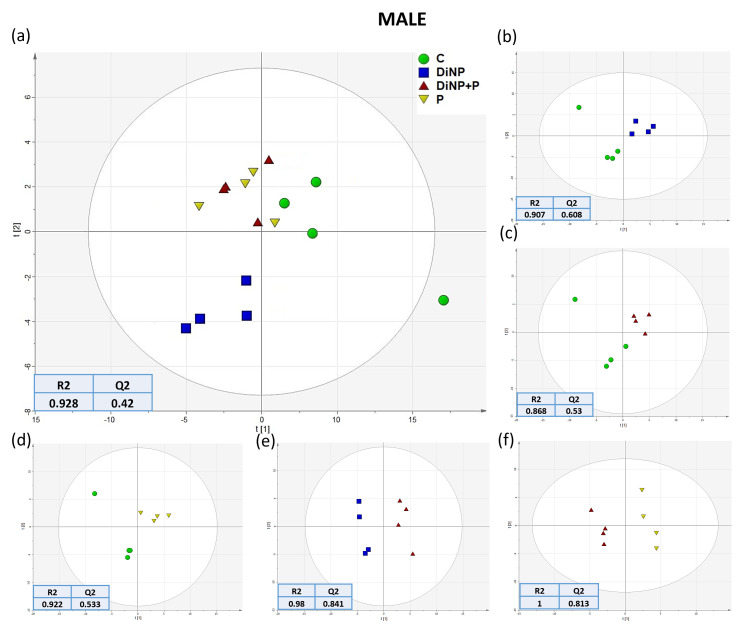
PLS-DA score plot analysis of (**a**) all groups, (**b**) DiNP vs. C, (**c**) DiNP + P vs. C, (**d**) P vs. C, (**e**) DiNP + P vs. DiNP, and (**f**) DiNP + P vs. P comparisons in males (*n* = 4 per group) with C (green), DiNP (blue), DiNP + P (red), and P (yellow). Each symbol in the PLS-DA represents a pool of at least three livers. Quality parameters of the built models are reported for each comparison.

**Figure 3 ijms-25-11151-f003:**
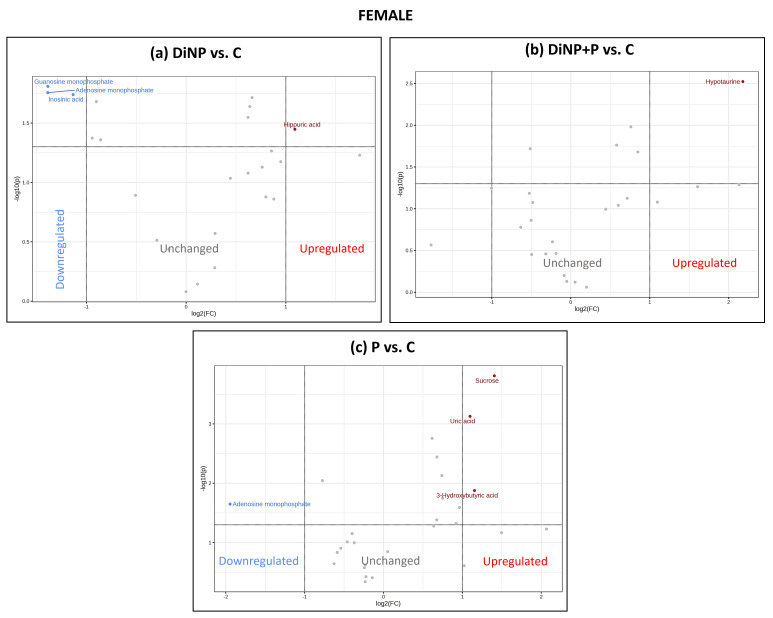
Volcano plot analysis of (**a**) DiNP vs. C, (**b**) DiNP + P vs. C, and (**c**) P vs. C in females. Up-regulated metabolites are shown in red, down-regulated metabolites are shown in blue, and not-significant metabolites are shown in gray.

**Figure 4 ijms-25-11151-f004:**
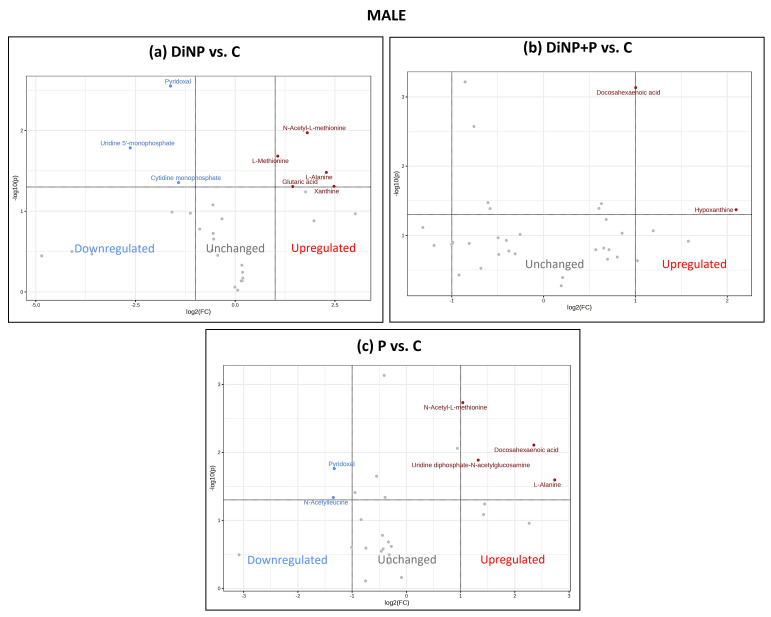
Volcano plot analysis of (**a**) DiNP vs. C, (**b**) DiNP + P vs. C, and (**c**) P vs. C in males. Up-regulated metabolites are shown in red, down-regulated metabolites are shown in blue, and not-significant metabolites are shown in gray.

**Figure 5 ijms-25-11151-f005:**
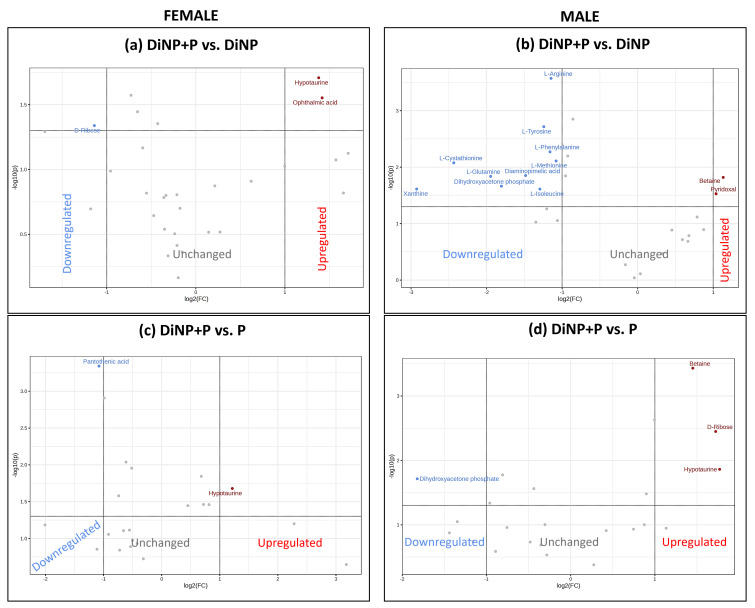
Volcano plots of DiNP + P vs. DiNP and DiNP + P vs. P in (**a**,**c**) females and (**b**,**d**) males. Up-regulated metabolites are shown in red, down-regulated metabolites are shown in blue, and not-significant metabolites are shown in gray.

**Figure 6 ijms-25-11151-f006:**
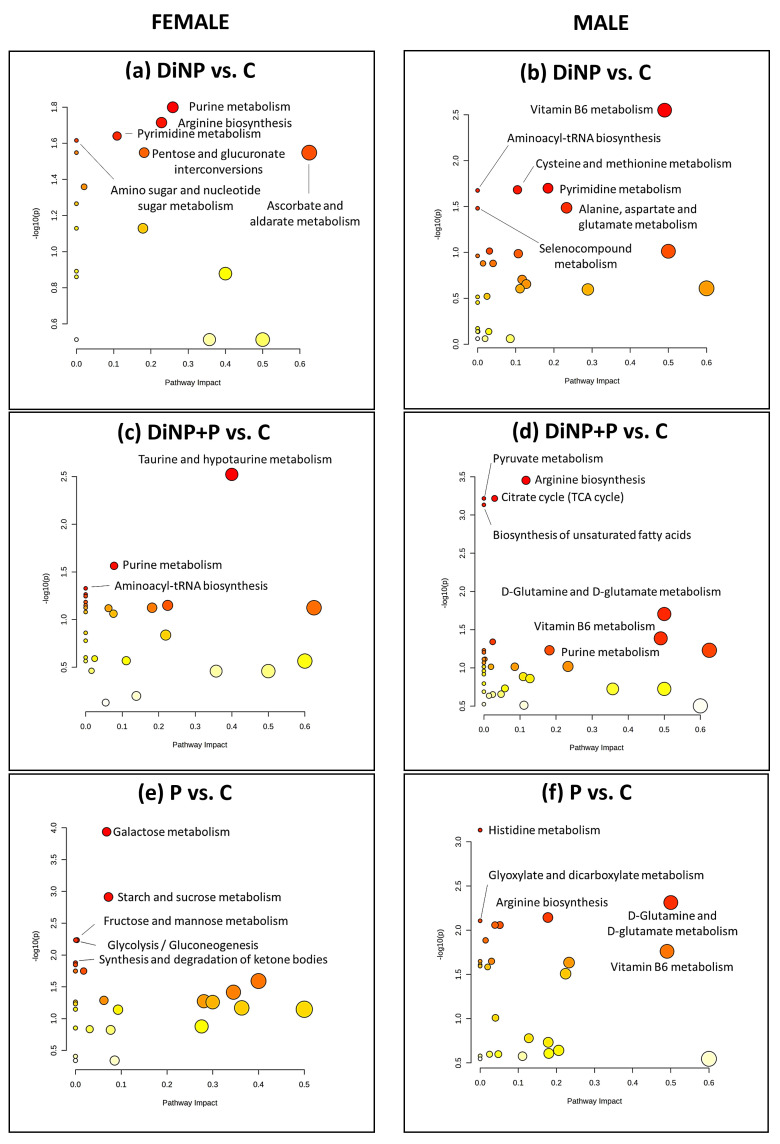
MetPA analysis of DiNP vs. C, DiNP + P vs. C, and P vs. C in (**a**,**c**,**e**) females and (**b**,**d**,**f**) males. The *X* axis stands for the relevance of the metabolite within the pathway, whereas the *Y* axis stands for the significance of the pathway in the comparison. The dot color represents the statistical significance level, with red being the most statistically significant and white being the least statistically significant. The size of the dots indicates the relevance of the metabolites inside that pathway: the bigger they are, the greater relevance they have.

**Figure 7 ijms-25-11151-f007:**
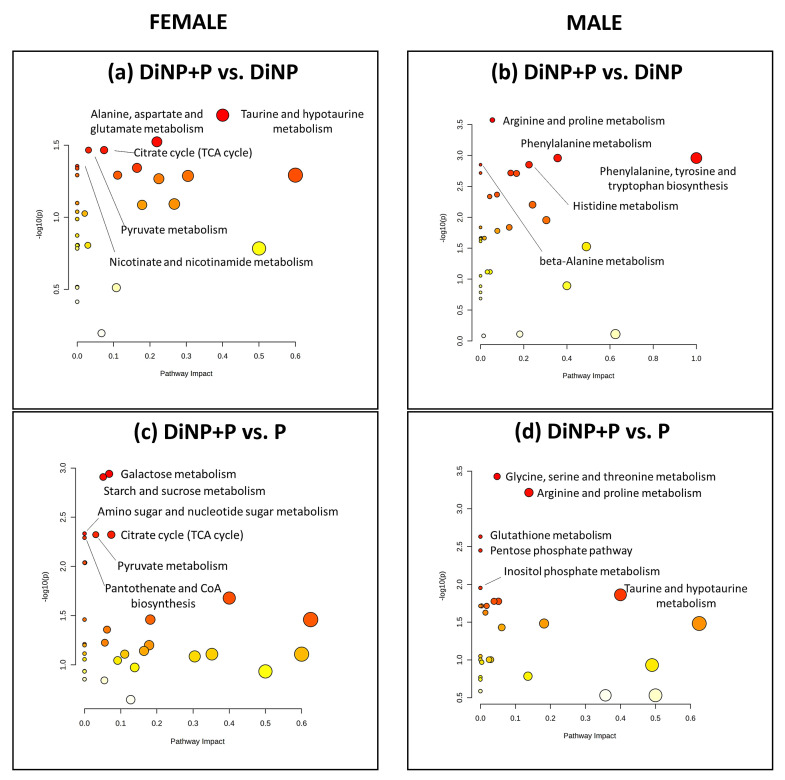
MetPA analysis of DiNP + P vs. DiNP and DiNP + P vs. P in (**a**,**c**) females and (**b**,**d**) males. The *X* axis stands for the relevance of the metabolite within the pathway, whereas the *Y* axis stands for the significance of the pathway in the comparison. The dots’ color represents the statistical significance level, with red being the most statistically significant and white the least statistically significant. The size of the dots indicates the relevance of the metabolites inside that pathway: the bigger they are, the greater relevance they have.

## Data Availability

Data will be provided on request.

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
