# Peer review of "Probiotics as Potential Tool to Mitigate Nucleotide Metabolism Alterations Induced by DiNP Dietary Exposure in Danio rerio"

_ijms, 2024, doi:10.3390/ijms252011151_

Round 1
Reviewer 1 Report
Comments and Suggestions for Authors
General comments
The present study aimed to investigate the potential hepatic sex-related toxicity of DiNP was investigated together with the mitigating action of the probiotic formulation SLAB51 in zebrafish adults by using a metabolomic approach. The result provides the first evidence of the disruptive actions of DiNP on hepatic nucleotide metabolism and mitigating action of the probiotic to reduce DiNP-induced response in a sex-related manner.
Just as the title " Probiotics as potential tool to mitigate the hepatic toxicity of DiNP in Danio rerio" means, you think that DiNP has hepatic toxicity to zebrafish, but you have no experimental evidence of hepatic toxicity of DiNP. Therefore, the authors should add some hepatic toxicity test of DiNP on zebrafish, for example, you can conduct an acute toxicity test on the fish to detect the hepatic transaminase and liver histopathology as to confirm that DiNP does has hepatotoxicity to zebrafish.
Additionally, oral exposure to DiNP and Probiotics should first affect the gut and its gut flora, so why did the authors not consider gastrointestine toxicity? I suggested that the authors can add the toxic effects of DiNP on the intestinal tract and intestinal microbiota, and how Probiotics SLAB51 has protective effects on the intestinal tract and microbiota.
Therefore, I suggest that the authors should add the above work and then the manuscript can be accepted.
Comments on the Quality of English LanguageMinor editing of English language required.
Author Response
Probiotics as potential tool to mitigate the nucleotide metabolism alterations induced by DiNP dietary exposure in Danio rerio
REVIEWER1
Open Review
(x) I would not like to sign my review report
( ) I would like to sign my review report
Quality of English Language
( ) I am not qualified to assess the quality of English in this paper.
( ) The English is very difficult to understand/incomprehensible.
( ) Extensive editing of English language required.
( ) Moderate editing of English language required.
(x) Minor editing of English language required.
( ) English language fine. No issues detected.
|
Yes |
Can be improved |
Must be improved |
Not applicable |
|
|
Does the introduction provide sufficient background and include all relevant references? |
(x) |
( ) |
( ) |
( ) |
|
Is the research design appropriate? |
( ) |
(x) |
( ) |
( ) |
|
Are the methods adequately described? |
( ) |
(x) |
( ) |
( ) |
|
Are the results clearly presented? |
( ) |
(x) |
( ) |
( ) |
|
Are the conclusions supported by the results? |
(x) |
( ) |
( ) |
( ) |
|
|
|
|
|
|
Comments and Suggestions for Authors
General comments
The present study aimed to investigate the potential hepatic sex-related toxicity of DiNP was investigated together with the mitigating action of the probiotic formulation SLAB51 in zebrafish adults by using a metabolomic approach. The result provides the first evidence of the disruptive actions of DiNP on hepatic nucleotide metabolism and mitigating action of the probiotic to reduce DiNP-induced response in a sex-related manner.
The authors thank the reviewer for the valuable comments provided to improve the manuscript. A point-by-point response to all the comments raised can be found below, written in red, and the corresponding corrections will be made in the manuscript, also highlighted in red to indicate the revised sections.
Just as the title " Probiotics as potential tool to mitigate the hepatic toxicity of DiNP in Danio rerio" means, you think that DiNP has hepatic toxicity to zebrafish, but you have no experimental evidence of hepatic toxicity of DiNP. Therefore, the authors should add some hepatic toxicity test of DiNP on zebrafish, for example, you can conduct an acute toxicity test on the fish to detect the hepatic transaminase and liver histopathology as to confirm that DiNP does has hepatotoxicity to zebrafish.
The authors would like to thank the reviewer for the valuable comment. In response, the title of the manuscript has been revised, replacing the term "toxicity" with "nucleotide metabolism alteration," which more accurately reflects the results presented. This correction has also been applied throughout the manuscript where applicable.
Regarding hepatic toxicity assessment, previous studies from our laboratory (10.1210/en.2017-00458; 10.1007/s00204-019-02494-7; 10.1016/j.envint.2018.06.011; 10.3389/FENDO.2018.00654/XML/NLM) using both zebrafish and seabream as experimental models, as well as studies from other laboratories using mammalian models (10.1007/s002040050634; 10.1016/j.scitotenv.2020.143631; 10.1016/j.fct.2014.03.027; 10.1177/0748233719900861; 10.1177/074823378700300204; 10.1093/EEP/DVAA017), have demonstrated the hepatic toxicity of DiNP. These studies indicate that DiNP impacts lipid metabolism, leading to liver steatosis and glycogen metabolism alterations, as well as liver proliferation, DNA synthesis, and oxidative stress. Furthermore, DiNP has been shown to act as a peroxisome proliferator, contributing to the onset of hepatocarcinoma (10.1016/S0300-483X(03)00260-9; 10.1007/S002040050634).
In light of the existing literature, the aim of the present study was to investigate whether SLAB51 administration could mitigate DiNP-induced hepatic disturbances, as we have previously demonstrated in the case of BPA exposure (10.1016/j.scitotenv.2023.169303). The metabolomic results offer a snapshot of the key metabolic changes occurring in the liver.
Additionally, oral exposure to DiNP and Probiotics should first affect the gut and its gut flora, so why did the authors not consider gastrointestine toxicity? I suggested that the authors can add the toxic effects of DiNP on the intestinal tract and intestinal microbiota, and how Probiotics SLAB51 has protective effects on the intestinal tract and microbiota.
Authors wish to thank the Reviewer for this valuable comment. As stated by the Reviewer, both the contaminant and the probiotic should affect the gut and its flora upon ingestion and possibly in turn hepatic metabolism could be affected. We are aware about the importance of having some evidence from the gut/intestine, but at the end of the trial these tissues were not sampled, thus to meet up the Reviewer comment, a Limitation section has now been added to the final part of the Discussion section (Line 363-370) in order to describe the here mentioned limitations and to indicate possible future research directions to widen the knowledge regarding this issue.
Therefore, I suggest that the authors should add the above work and then the manuscript can be accepted.
Reviewer 2 Report
Comments and Suggestions for Authors
The authors provide evidence that probiotics might counteract DiNP toxicity in an animal model.
The paper is clearly written. However, the discussion could be expanded considering the following aspects:
- what is the relevance of the animal model and studied effects for humans. Are the pathways active in humans?
- the conclusion suggests that EFSA should re-evaluate DiNP. However, this demand could be made more specific also including suggestions for further research that would allow EFSA to make a re-evaluation. I believe this single study would not convince EFSA to do so.
- the sex specificity is a bit puzzling. Typically, liver effects are not sex-specific, besides perhaps in magnitude due to capacity for detoxificiation (e.g. compare alcohol). Can the alternative hypothesis be explored in the limitations sections, e.g. that the sex specificity might be an artifiact of analytical errors, low sample size etc
Some further specific remarks:
Line 19: explain SLAB51
Line 20, 60 and throughout: "kg" symbol must be lower case! (SI unit)
Lines 230-235 and possibly throughout: the names of chemicals must be lower case in English
Author Response
Probiotics as potential tool to mitigate the nucleotide metabolism alterations induced by DiNP dietary exposure in Danio rerio
REVIEWER 2
Open Review
(x) I would not like to sign my review report
( ) I would like to sign my review report
Quality of English Language
( ) I am not qualified to assess the quality of English in this paper.
( ) The English is very difficult to understand/incomprehensible.
( ) Extensive editing of English language required.
( ) Moderate editing of English language required.
( ) Minor editing of English language required.
(x) English language fine. No issues detected.
|
Yes |
Can be improved |
Must be improved |
Not applicable |
|
|
Does the introduction provide sufficient background and include all relevant references? |
(x) |
( ) |
( ) |
( ) |
|
Is the research design appropriate? |
(x) |
( ) |
( ) |
( ) |
|
Are the methods adequately described? |
(x) |
( ) |
( ) |
( ) |
|
Are the results clearly presented? |
(x) |
( ) |
( ) |
( ) |
|
Are the conclusions supported by the results? |
(x) |
( ) |
( ) |
( ) |
Comments and Suggestions for Authors
The authors provide evidence that probiotics might counteract DiNP toxicity in an animal model.
The paper is clearly written. However, the discussion could be expanded considering the following aspects:
The authors thank the reviewer for the appreciation on the manuscript and for the valuable comments provided to improve it. A point-by-point response to all the comments raised can be found below, written in red, and the corresponding corrections will be made in the manuscript, also highlighted in red to indicate the revised sections.
- what is the relevance of the animal model and studied effects for humans. Are the pathways active in humans?
The authors would like to thank the reviewer for this important question, which allows us to elaborate on the relevance of the chosen animal model and the significance of the outcomes presented in this study. Over the past few decades, zebrafish has emerged as an important animal model for biomedical research (10.1093/af/vfz020; 10.1016/j.heliyon.2023.e14557; 10.1203/PDR.0b013e318186e609; 10.5772/intechopen.91319; 10.3390/ijms221910766). Approximately 70% of human genes have at least one clear zebrafish orthologue (10.1038/nature12111). Zebrafish is also a valuable model for toxicological studies (10.1016/j.ecoenv.2024.116023; 10.1093/toxsci/kfy044; 10.1016/j.biopha.2024.116160; 10.1016/bs.ctdb.2016.10.007; 10.1093/toxsci/kfi110; 10.20517/2572-8180.2017.15; 10.1007/s11356-015-5362-1), particularly due to the similarity in liver function (10.3390/cells12182246; 10.1053/j.gastro.2015.08.034; 10.1016/j.ddmod.2017.04.003; 10.37349/edd.2023.00017) and other organs (10.3390/biomedicines12030693; 10.1016/j.nbd.2010.05.010; 10.1038/s41420-018-0109-7) between humans and Danio rerio, as well as their comparable responses to toxicant exposure.
Given this evidence, using zebrafish to study the potential hepatic metabolic disturbances induced by dietary DiNP exposure and the possible mitigating effects of SLAB51 allows for easier translation of these findings to humans. Regarding the pathways investigated in this study through metabolomics, both purine and pyrimidine metabolisms are active and highly conserved between zebrafish and humans (10.1016/j.tibs.2016.09.009; 10.3390/ijms24087027; 10.3389/fimmu.2018.01697; 10.1016/j.molmet.2020.02.005; 10.3389/fonc.2021.684961; 10.1080/15257770.2023.2298742).
A brief description of this information has been added to the manuscript in the discussion section, specifically at the beginning (Lines 250–254) regarding the importance of zebrafish as a model for human research, and in the discussion on purine and pyrimidine metabolism (Lines 2634–266).
- the conclusion suggests that EFSA should re-evaluate DiNP. However, this demand could be made more specific also including suggestions for further research that would allow EFSA to make a re-evaluation. I believe this single study would not convince EFSA to do so.
Thank you for your comment. The final part of the conclusion section has been revised to position this study as a starting point and to include suggestions for future research that could assist EFSA and policymakers in re-evaluating the DiNP limits and making decisions regarding this contaminant. The revision emphasizes that only by thoroughly characterizing the toxicity of this contaminant can an informed decision be made. The revised section can be found in the manuscript in red, between lines 472–475.
- the sex specificity is a bit puzzling. Typically, liver effects are not sex-specific, besides perhaps in magnitude due to capacity for detoxificiation (e.g. compare alcohol). Can the alternative hypothesis be explored in the limitations sections, e.g. that the sex specificity might be an artifiact of analytical errors, low sample size etc
A recent study has clearly demonstrated the sex-specific nature of zebrafish hepatic metabolism (10.1371/journal.pone.0053562). The results show a significant enrichment of vitellogenin transcripts in the livers of spawning females (80% of total RNA), along with pathways related to ribosome/translation, estrogen signaling, and lipid transport. In contrast, males showed enrichment in pathways involved in oxidation-reduction, carbohydrate metabolism, coagulation, and protein transport and localization. Additionally, previous studies from our laboratory have shown that contaminants can impact hepatic metabolic function in a sex-specific manner in adult zebrafish exposed to the herbicide Glyphosate (10.3390/ijms23052724) or the plasticizer Bisphenol A (10.1016/j.scitotenv.2023.169303). Our findings align with studies from other laboratories, which demonstrate the ability of Bisphenol A (10.1016/j.cbpc.2023.109616) and Polychlorinated Biphenyls (10.1016/j.chemosphere.2019.01.094) to disrupt hepatic lipid metabolism in a sex-specific manner in zebrafish. Likewise, exposure to Perfluorooctanoic acid has been shown to sex-specifically affect the transcriptional expression of FABPs in zebrafish liver (10.1021/es300147w), and Perfluorobutanesulfonate exposure has been reported to alter hepatic lipid metabolism in a sex-specific way (10.1021/acs.est.0c02345). Collectively, these studies highlight the sex-specific nature of zebrafish hepatic responses both under normal conditions and in response to pollutant exposure. The data reported in the present study are consistent with this body of literature. To better clarify the sex-specific nature of the observed responses to DiNP, SLAB51, and their combination, the information provided in this response has been incorporated into the discussion section of the manuscript in lines 352–355 and lines 358–359.
Some further specific remarks:
Line 19: explain SLAB51
An explanation for the use of the probiotic formulation SLAB51 in this trial has been added to the main text of the manuscript, highlighted in red in the abstract section, lines 20–21.
Line 20, 60 and throughout: "kg" symbol must be lower case! (SI unit)
According to the comment, the kg symbol was revised throughout the manuscript and the corrections are marked in red.
Lines 230-235 and possibly throughout: the names of chemicals must be lower case in English
In response to the comment, the chemical names have been revised and written in lowercase throughout the manuscript, with the changes highlighted in red.
Reviewer 3 Report
Comments and Suggestions for Authors
Regarding the suitability of DiNP, which is currently used as an alternative to DEHP, the authors showed that despite the tightening of regulations by the European Food Safety Authority (EFSA), it exhibits metabolic toxicity in the liver, including gender differences, and demonstrated that this is mitigated by probiotics. The paper is well organized, with an introduction, methods, results, and discussion, and an analysis of a large amount of data.
As a reviewer, I believe that the following two improvements would help readers understand the paper better.
1) Results: If there is data on tissue damage to the liver, please add it to the results. The authors have published a paper in which they analyzed the toxicity of BPA in organs other than the liver using zebrafish, and showed that the same probiotic (SLAB51) had a toxicity-mitigating effect. Did they not examine data on tissue damage to the liver in this paper? If not, please explain why.
2) Discussion: Please add a paragraph on limitations of this paper to the discussion. The authors have developed inferences based on the toxicity data of DiNP, including sex differences in the liver's metabolic system, citing appropriate literature. This point is commendable, but since further research is necessary to prove these hypotheses, we request that this be added as a limitation to this paper.
Author Response
Probiotics as potential tool to mitigate the nucleotide metabolism alterations induced by DiNP dietary exposure in Danio rerio
REVIEWER 3
Open Review
( ) I would not like to sign my review report
(x) I would like to sign my review report
Quality of English Language
( ) I am not qualified to assess the quality of English in this paper.
( ) The English is very difficult to understand/incomprehensible.
( ) Extensive editing of English language required.
( ) Moderate editing of English language required.
( ) Minor editing of English language required.
|
Yes |
Can be improved |
Must be improved |
Not applicable |
|
(x) |
( ) |
( ) |
( ) |
|
( ) |
(x) |
( ) |
( ) |
|
( ) |
(x) |
( ) |
( ) |
|
( ) |
(x) |
( ) |
( ) |
|
( ) |
(x) |
( ) |
( ) |
(x) English language fine. No issues detected.
Does the introduction provide sufficient background and include all relevant references?
Is the research design appropriate?
Are the methods adequately described?
Are the results clearly presented?
Are the conclusions supported by the results?
Comments and Suggestions for Authors
Regarding the suitability of DiNP, which is currently used as an alternative to DEHP, the authors showed that despite the tightening of regulations by the European Food Safety Authority (EFSA), it exhibits metabolic toxicity in the liver, including gender differences, and demonstrated that this is mitigated by probiotics. The paper is well organized, with an introduction, methods, results, and discussion, and an analysis of a large amount of data.
The authors thank the reviewer for the appreciation on the manuscript and for the valuable comments provided to improve it. A point-by-point response to all the comments raised can be found below, written in red, and the corresponding corrections will be made in the manuscript, also highlighted in red to indicate the revised sections.
As a reviewer, I believe that the following two improvements would help readers understand the paper better.
1) Results: If there is data on tissue damage to the liver, please add it to the results. The authors have published a paper in which they analyzed the toxicity of BPA in organs other than the liver using zebrafish, and showed that the same probiotic (SLAB51) had a toxicity-mitigating effect. Did they not examine data on tissue damage to the liver in this paper? If not, please explain why.
The authors would like to thank the reviewer for the valuable comment. Regarding hepatic toxicity assessment, previous studies from our laboratory (10.1210/en.2017-00458; 10.1007/s00204-019-02494-7; 10.1016/j.envint.2018.06.011; 10.3389/FENDO.2018.00654/XML/NLM) using both zebrafish and seabream as experimental models, as well as studies from other laboratories using mammalian models (10.1007/s002040050634; 10.1016/j.scitotenv.2020.143631; 10.1016/j.fct.2014.03.027; 10.1177/0748233719900861; 10.1177/074823378700300204; 10.1093/EEP/DVAA017), have demonstrated the hepatic toxicity of DiNP. These studies indicate that DiNP impacts lipid metabolism, leading to liver steatosis and glycogen metabolism alterations, as well as liver proliferation, DNA synthesis, and oxidative stress. Furthermore, DiNP has been shown to act as a peroxisome proliferator, contributing to the onset of hepatocarcinoma (10.1016/S0300-483X(03)00260-9; 10.1007/S002040050634).
In light of the existing literature, the aim of the present study was to investigate whether SLAB51 administration could mitigate DiNP-induced hepatic disturbances, as we have previously demonstrated in the case of BPA exposure (10.1016/j.scitotenv.2023.169303). The metabolomic results offer a snapshot of the key metabolic changes occurring in the liver.
Unfortunately, at the end of the trial, tissues other than liver were not sampled and since the amount of liver requested for metabolomic analysis was high and required pooling at least 3 livers together to reach the necessary amount for extraction, was not possible to sample other livers for additional analyses. Thus to meet up the Reviewer comment, a Limitation section has now been added to the final part of the Discussion section (Line 363-370) in order to describe the here mentioned limitations and to indicate possible future research directions to widen the knowledge regarding this issue.
2) Discussion: Please add a paragraph on limitations of this paper to the discussion. The authors have developed inferences based on the toxicity data of DiNP, including sex differences in the liver's metabolic system, citing appropriate literature. This point is commendable, but since further research is necessary to prove these hypotheses, we request that this be added as a limitation to this paper.
Thank you for your comment. A limitation section has been added describing the study limitations, indicating this paper as a starting point and identifying possible future directions for research investigations in this field.
Round 2
Reviewer 1 Report
Comments and Suggestions for Authors
NONE.
Comments on the Quality of English LanguageGOOD.